# A Standardized Procedure to Build a Spectral Library for Hazardous Chemicals Mixed in River Flow Using Hyperspectral Image

**Yeonghwa Gwon [1], Dongsu Kim [1,*], Hojun You [2], Su-Han Nam [3] and Young Do Kim [3]**

1  Department of Civil & Environmental Engineering, Dankook University, Yongin-si 16890, Geyonggi-do, Republic of Korea
2  K-water Institute, Daejeon-si 34045, South Chungcheong, Republic of Korea
3  Department of Civil & Environmental Engineering, Myongji University, Yongin-si 17058, Geyonggi-do, Republic of Korea
*  Correspondence: dongsu-kim@dankook.ac.kr; Tel.: +82-31-8005-3611

**Abstract:** The occurrence of natural disasters as a consequence of accidental hazardous chemical spills remains a concern. The inadequate, or delayed, initial response may fail to mitigate their impact; hence, imminent monitoring of responses in the initial stage is critical. Classical contact-type measurement methods, however, sometimes miss solvent chemicals and invoke risks for operators during field operation. Remote sensing methods are an alternative method as non-contact, spatially distributable, efficient and continuously operable features. Herein, we tackle challenges posed by the increasingly available UAV-based hyperspect ral images in riverine environments to identify the presence of hazardous chemical solvents in rivers, which are less investigated in the absence of direct measurement strategies. We propose a referable standard procedure for a unique spectral library based on pre-scanning hyperspectral sensors with respect to representative hazardous chemicals registered on the national hazardous chemical list. We utilized the hyperspectral images to identify 18 types of hazardous chemicals injected into the river in an outdoor environment, where a dedicated hyperspectral ground imaging system mounted with a hyperspectral camera was designed and applied. Finally, we tested the efficiency of the library to recognize unknown chemicals, which showed >70% success rate.

**Keywords:** hyperspectral; hazardous chemicals; spectral library; chemical accident; recognition test

## 1. Introduction

There are increasing social concerns regarding water pollution and natural disaster attributed to hazardous chemicals spills. With industrialization, accidental hazardous chemical spills in riverine environments occur more frequently, which has adverse effects on humans and the ecological environment [1]. The inadequate or delayed initial response may lead to serious environmental disasters and fail to mitigate their impact; thus, appropriate actions and responses in the initial stages are critical. However, the leakage of chemicals into rivers is particularly challenging to detect as they are mostly transparent solvents. For example, when transparent sodium hypochlorite was leaked after an accidental damage to a storage tank filled with a detergent on 11 May 2018, the leaked chemical flowed into Gam stream and caused significant harm to aquatic life, such as fish [2]. Local authorities were able to recognize the type of chemical and its leaked location by indirectly tracking a fire in a nearby factory. No detecting protocol worked, while the chemical was never detected in the middle of the spilling accident. The imminent sensing of solvent chemical leaked in the river has been difficult to perform since identifying the chemical occurred only after detecting the certain suspicious spill in rivers where "direct" and "contact" in-situ detectors were conventionally applied as subsequent sensing devices, such as pH

paper, simple identification and detection kits or detector tube devices, as well as electronic detectors, such as a portable ion spectrometer or portable Fourier transform infrared (FT-IR) spectroscope. These contact-type measurement methods, however, may not be able to detect the presence of chemicals due to sensor's local positioning and contact nature when spilled chemicals are disseminated spatially biased on opposite sides of a river cross-section. Also, this type of direct sensing invokes risks for field operators while performing system installation, measurement and checking status of toxic chemicals. Therefore, there have been increasing needs for the development of alternatively efficient, spatially distributable and safe technologies enabling seamless monitoring against hazardous chemicals spilled in river water. Among various alternative ways, remote sensing is a potential candidate when considering its advantages such as minimal contact, spatial distribution, efficiency and ease of operation. Recent advances in platforms, such as CCTV and UAV, made remote sensing more plausible for this kind of river monitoring.

From such a perspective, the present study specifically proposes a remote sensing method of using hyperspectral images to identify the type of hazardous chemicals, where spectral images are continuously monitored along a cross-section of river. More specifically, we developed a dedicated spectral library of solvent hazardous chemicals that helped to characterize specific features of a given spectrum. Already, hyperspectral images have been increasingly applied for assessing the hydraulic and ecological characteristics of rivers. For example, monitoring river water quality (e.g., Chl-a, TSS and turbidity) using aerial and satellite-based hyperspectral images has been developed [3–7]. River depth measurements based on drone-based hyperspectral images are being actively investigated [8]. The present study expands their applications for detecting and identifying dissolved or floating chemicals spilled in a river using a hyperspectral image [9–11]. The motivation for using hyperspectral images for chemical identification is to construct a unique spectral library for each chemical. In fact, spectral libraries have played a key role in the identification of terrain objects, such as minerals, soils, vegetation, artifacts and liquids [12], when spectral images began to be available following the era of satellites. However, there have been very few spectral libraries researched for substances based on solvents, such as chemicals mixed in river water, mainly due to the fact that spatial resolution of conventional spectral image from conventional satellites supporting spectral images (i.e., 30 m) is not suitable to detect chemicals spilled in relatively small rivers. Furthermore, existing spectral libraries for identification of substances mixed in water, such as those developed by USGS [13], Johns Hopkins University [14], NASA [14] and the ECOSTRESS spectral library version 1.0, are mostly unable to detect non-chemical materials, such as frost, ice, distilled water, seafoam, seawater, coarse granular snow and red-coated algae water [15]. Additionally, these studies have mainly constructed spectral libraries based on indoor spectrometers while missing critical signatures for water solvent, light-transmittable substances applicable to natural rivers under sunlight.

Considering that a reliable spectral library is enough to utilize hyperspectral images acquired from river environment for monitoring hazardous chemicals, very few prior studied spectral libraries are available. In this context, this study helps to derive a spectral library for identifying various hazardous chemicals floating in or mixed in river water. Several issues must be prioritized before newly streamlining a spectral library, where the hyperspectral images acquired from platforms, such as UAVs, raise new issues in contrast to those obtained from former spectral investigations conducted in controlled indoor conditions. First, when building a unique chemical spectral library, the spectral characteristics of river water must be removed as it exhibits higher reflective properties than the dissolved chemicals. Also, the spectral characteristics for each river water may differ. The spectral impact from the surrounding solvent should be discriminated and eliminated to isolate pure spectrum from the chemicals. Second, unlike with point spectroradiometer, when building a standard spectral library from hyperspectral images recorded in-plane units, post-processing and averaging of multiple spectral signatures are required.

Thus, outlining the procedure for averaging multiple spectral information for targeted chemicals with outlier removal to obtain a representative spectrum as the unique spectral trend for further versatile usages is essential. Third, chemicals dissolved in river water with the same properties exhibit, to some extent, different spectral characteristics depending on field observation conditions, i.e., solar intensity and angle. Thus, conversion to reflectance requires standard quantification through radiometric calibration using multiple reflection tarps with known absolute reflectance. Fourth, despite similar spectral patterns, the same chemicals may show different reflectance levels depending on the chemical concentration and solar intensity. Hence, the reflectance levels must be standardized to avoid challenges in future identification. Consequently, in addition to the existing method for building a spectral library, a new standard procedure for building a unique spectral library for each chemical should be established. This spectral library will allow the identification of dissolved chemicals, in river water, from UAV-based hyperspectral images. The core contribution of this study is the proposed standard procedure and method for building a spectral library.

From the above perspective, we tackle the challenges posed by the increasingly available UAV-based hyperspectral images in riverine environments to identify the presence of hazardous chemical solvents in rivers which have been less investigated in the absence of direct measurement. We propose a referable standard procedure for constructing a unique spectral library based on pre-scanning hyperspectral sensors mounted on a drone with respect to representative hazardous chemicals registered on the national hazardous chemical list. The spectral library for the chemicals was built on the condition that the chemicals are basically mixed with natural rivers in the form of dissolved or floating ions on river water. We utilized the hyperspectral images for 18 types of hazardous chemicals injected into river water in an outdoor environment, where a dedicated hyperspectral ground imaging system mounted with a hyperspectral camera was designed and applied.

## 2. Materials and Methods

### 2.1. Target River and Hazardous Chemicals

The hazardous chemicals supplied to build a unique spectral library are close to undiluted solutions of high concentrations. To build a spectral library of hazardous chemicals mixed in river water, we collected water from Gam Stream of the Nakdong River water system located in the southern part of South Korea. There are industrial complexes located upstream of the Gam Stream, where several former spill incidents of hazardous chemicals occurred, such as the detection of perchlorate at major water intake sites on the Nakdong River in July 2006; a phenol spill due to a fire accident in the Gimcheon site of Kolon Industries in March 2008; and the temporary suspension of water intake due to phenol detection in Gumi Metropolitan Water Treatment Plant. Thus, the Gam Stream was selected as a testing subject in this study since there is a real concern regarding water quality due to chemical spills in this area.

As mentioned, 18 types of chemicals were selected as hazardous chemicals, consisting of five types of organic matter and 13 types of inorganic matter (Table 1). By pH, the hazardous chemical samples of 3000 ppm consisted of 8 types of acidic chemicals, seven types of neutral chemicals and 3 types of basic chemicals. The hazardous chemicals were turned into 0.5 M stoke solutions by diluting in distilled water, considering the molecular weight and purity of each chemical. Samples were produced by diluting these 0.5 M stoke solutions in river water. The samples prepared to capture hyperspectral images were placed in a wide shallow container with a glass cover to secure stability and minimize light diffraction that occurs on a spherical surface. Handling hazardous chemicals that have no cover is challenging; therefore, 250 mL samples were collected in covered crystallizing dishes with a diameter of 80 mm and a height of 45 mm. Hyperspectral images of samples in these containers were captured.

**Table 1.** Hazardous chemicals used to build a spectral library.

| No. | CAS No. | Name | Chemical Formula | Color | Organic | pH |
|---|---|---|---|---|---|---|
| 1 | 107-07-3 | 2-Chloroethanol | $C_2H_5ClO$ | Colorless | Organic | 7.42 |
| 2 | 556-52-5 | Glycidol | $C_3H_8O_2$ | Colorless | Organic | 8.29 |
| 3 | 78-93-3 | Methyl ethyl ketone | $C_4H_8O$ | Colorless | Organic | 8.22 |
| 4 | 7664-39-3 | Hydrogen fluoride | HF | Colorless | Inorganic | 2.18 |
| 5 | 7726-95-6 | Bromine | $Br_2$ | Dark reddish-brown, Dark red | Inorganic | 5.8 |
| 6 | 7784-34-1 | Arsenic trichloride | $AsCl_3$ | Colorless, Yellow oily fuming liquid | Inorganic | 6.69 |
| 7 | 7719-12-2 | Phosphorus trichloride | $PCl_3$ | Colorless, Yellow oily fuming liquid | Inorganic | 1.53 |
| 8 | 143-33-9 | Sodium cyanide | NaCN | White crystalline solid, Colorless liquid | Inorganic | 10.90 |
| 9 | 7664-41-7 | Ammonia | $NH_3$ | Colorless liquid | Inorganic | 10.47 |
| 10 | 7647-01-0 | Hydrogen chloride | HCl | Colorless | Inorganic | 1.05 |
| 11 | 869-24-9 | 2-Chloroethyldiethylammonium chloride | $C_6H_{14}ClN \cdot HCl$ | Colorless liquid | Organic | 6.50 |
| 12 | 7719-09-7 | Thionyl chloride | $SOCl_2$ | Colorless to yellow to reddish liquid | Inorganic | 1.41 |
| 13 | 10025-87-3 | Phosphorus oxychloride | $POCl_3$ | Colorless to yellow, oily liquid | Inorganic | 1.46 |
| 14 | 1341-49-7 | Ammonium bifluoride | $(NH_4)HF_2$ | White crystals, Colorless liquid | Inorganic | 3.49 |
| 15 | 108-88-3 | Toluene | $C_7H_8$ | Colorless | Organic | (basic) |
| 16 | 7681-49-4 | Sodium fluoride | NaF | White powder or colorless crystals, colorless liquid | Inorganic | 6.85 |
| 17 | 7789-23-3 | Potassium fluoride | KF | White crystalline, colorless liquid | Inorganic | 7.27 |
| 18 | 7664-93-9 | Sulfuric acid | $H_2SO_4$ | Colorless to dark-brown, oily liquid | Inorganic | 1.34 |

## 2.2. Hyperspectral Image Collection

The hyperspectral sensors are generally passive types that collect light reflected from objects after they absorb sunlight. The hyperspectral sensor used in this study was Corning's microHSI 410 SHARK, as shown in Figure 1a [16]. This hyperspectral sensor has a wavelength range of VNIR, i.e., 400–1000 nm, a spectral resolution of 4 nm, and provides a total of 150 spectral bands. The sensor size, including the lens, is 13.6 cm × 8.7 cm × 7.0 cm, with a weight of 0.68 kg. Since the sensor is lightweight, it can be easily mounted on UAVs. The sensor collects the spectral information using the line scanning method with a push-broom that spatially records the light passing through a prism [17]. Basically, spectral outliers were removed using a Kalman filter. The observation angle is 29.5°, while 682 pixels are collected per scanning line at a frequency of 300 Hz. Hyperspectral images captured using the line scanning technology enable the measurement of space as the spectral characteristics collected at each observation are stored linearly. The linearly stored spectral characteristics along the traveling path of the platform were recorded (Figure 1b).

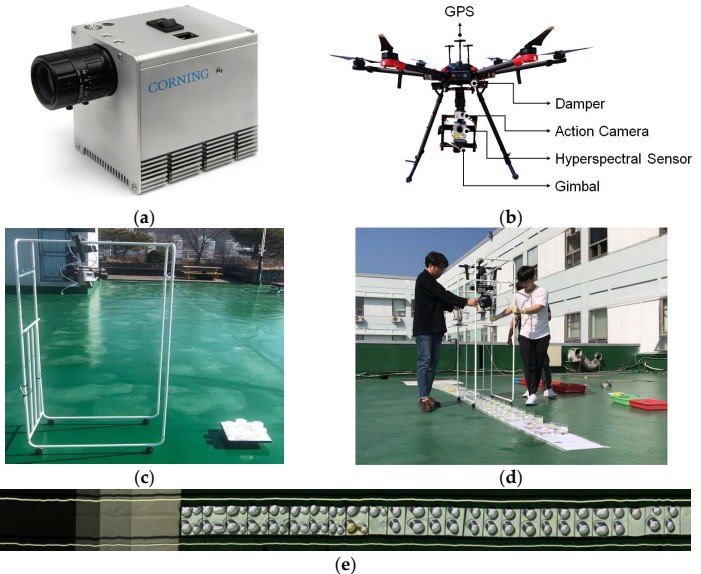

**Figure 1.** Hyperspectral image sensor and its ground-based outdoor portable holder to obtain spectral images for various hazardous chemicals. (**a**) Hyperspectral sensor used for building spectral image; (**b**) a drone system with hyperspectral sensor, GPS and gimbal supposed to monitor hazardous chemicals in riverine environments; (**c**) a portable holder mounted with hyperspectral sensor on the ground and chemical samples to be photographed; (**d**) manual mobile operation of the portable holder on top of biker containing chemicals; (**e**) resultant hyperspectral image where prepared samples of hazardous chemical were monitored concurrently.

As seen in Figure 1c, the hazardous chemical samples were captured with a hyperspectral camera mounted on a portable holder at a height of approximately 1.4 m to enable portable photography according to the line scanning method for collecting various samples concurrently. By slowly migrating the holder with a hyperspectral camera at a constant speed, various chemical samples could be collected almost simultaneously under natural light conditions (Figure 1d,e). Due to the pre-scanning method, the hyperspectral camera mounted on a UAV or a ground holder is sensitive to wind, flight vibrations and observation angle. The vibration of the mounted drone and its portable holder should be considered, which occurs when the flight path is changed compared to a relatively stable manned airplane. Thus, it was challenging to generate hyperspectral images without proper post-processing since line scanned images could not be properly combined. Consequently, clear hyperspectral images were often not obtained due to the failure of proper image registration. To cope with this issue, the installation of a gimbal was indispensable to minimize and correct shaking mathematically. In this study, hyperspectral raw data with minimal shaking could be obtained by applying DJI's RONIN-MX (Figure 1b,c); the results are shown in Figure 1e. Further, white A4 sheets were placed below the samples to be captured, for minimizing the effects of the floor material.

The hyperspectral image depicted in Figure 1e was captured at 1:30 pm on 16 October 2019, when the solar angle elevation was very high. The temperature was 15.3 °C, and the relative humidity was 72.6%. The solar azimuth was 215°53′18.25″, and its altitude was 38°5 4′12.9″.

Notably, the reflectance, a normalized spectral signal, mainly utilized to build a library or to conduct further analysis, is obtained by dividing the intensity of the light source and the intensity of the reflected light from objects. A conventional ground-based spectrometer and spectroradiometer can calculate relatively accurate reflectance without further processing.

However, the line scan-based hyperspectral sensor stores the light intensity in the form of a radiance and, hence, should be converted into reflectance in the post-process with respect to an absolute reference, such as a white plate (i.e., a standard reference panel (Spectralon (Labsphere, North Sutton, NH, USA)) with 99.9% reflectance applied where the reflectance derived from radiance can be expressed in Equation (1). When single reference like white plate is not sufficient, a couple of reference plates with different reflectance values can be alternatively applied. This study applied the latter approach that will be explained later on.

$$\text{Reflectance }(\lambda) = \frac{\text{Radiance }(\lambda)}{\text{Radiance}_{\text{white plate}}(\lambda)} \times 100 \tag{1}$$

Figure 2b shows the raw spectral signature expressed as radiance extracted from 100 pixels at the center of the container in the photographed chemicals (Figure 2a). Although there are subtle differences between all chemicals, the differences between chemicals mixed in river water are insignificant (Figure 2b–d). Furthermore, the usual spectral library acquired from the spectrometer (i.e., point and time-averaged) through artificial lights in the existing laboratory can be used to distinguish between chemicals without a separate standard processing procedure since it provides one spectral signature. However, as with the approach adopted here, multiple spectral signatures (e.g., 100 spectrums) could be acquired from one material using the line-scan hyperspectral imaging system. If the solvent, such as the river water, has a strong reflectance compared to the chemical, the utility of such a library, based on raw spectral signature (Figure 2), can be limited in future identification.

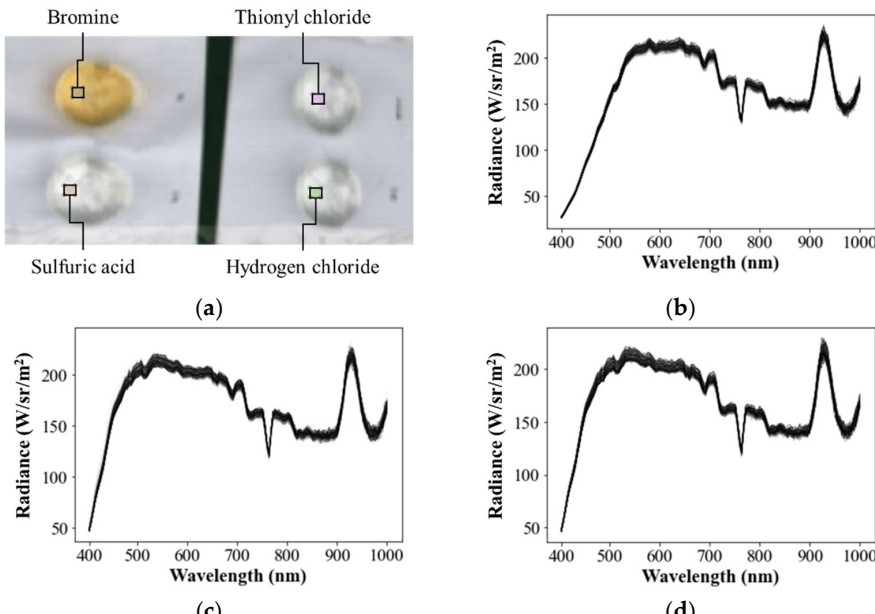

**Figure 2.** Example of spectral extraction and extracted spectral signatures. (**a**) Example of extraction of the spectral signature; (**b**) spectral signatures of river water and bromine mixture; (**c**) spectral signatures of river water and sulfuric acid mixture; (**d**) spectral signatures of river water and hydrogen chloride mixture.

## 3. Results and Discussion

### 3.1. Procedure for Building a Spectral Library

#### 3.1.1. Deriving Radiometric Calibration Equation

Radiometric calibration must be performed for the obtained hyperspectral images in order to remove the atmospheric absorption and scattering effect. Generally, this process involves the conversion of radiance to the reflectance as well as removing the atmospheric effects [18]. It is desirable to collect the coincident spectral radiation measurements (i.e., to the hyperspectral image acquisition) using a field spectroradiometer [19]. However, it may be challenging to approach the site during image acquisition or to operate the hyperspectral sensor and spectroradiometer concurrently. Therefore, we concluded that the collected spectral data (i.e., radiance) can be better converted into reflectance using a standard reference panel (Spectralon (Labsphere, North Sutton, NH, USA)) or calibration panel. Furthermore, since different signatures may be derived from the same chemical depending on the optical and equipment conditions at the time of hyperspectral imaging, reference values for converting the measurements (i.e., DN and radiance) to reflectance are required. For this purpose, the so called "reflectance tarps" made of specific materials maintaining standard stepwise reflectance values of 55, 44, 22 and 5% (Figure 3a) were applied using a calibration panel together with the hyperspectral images of hazardous chemicals. The stepwise reflectance tarps would guarantee better performance in conversion of radiance to reflectance rather than using a single white plate of 99%. Since the accurate reflectance values of the reflectance tarps are known and mostly fixed for any spectral bands as seen in Figure 3b, a reflectance-radiance relationship for each spectral band (Figure 3e) can be established by constructing a relational equation and standardizing imaging conditions for both the reflectance tarps and chemicals. Notably, the hyperspectral sensor (microHSI (Corning Advanced Optics, Corning, NY, USA)) provides raw data as DN (or radiance), which is the light intensity, for a total of 150 spectral bands in 4 nm intervals in the spectral range of 400–1000 nm. The actual reflectance values of the reflectance tarps are already set in all the 150 bands. For example, the 55% reflectance tarp shows 55% reflectance in all the wavelengths. In this study, we verified the reflectance values of the reflectance tarps using a spectroradiometer (PSR-2500 (Spectral Evolution Inc., Haverhill, MA, USA)). The

PSR-2500 spectroradiometer is a well-established and highly reliable passive device for collecting spectral information (Figure 3b). The results were highly similar to the values of reflectance tarps in most wavelengths except 400–420 nm. However, the spectral data of the reflectance tarps, collected using a hyperspectral sensor (microHSI (Corning Advanced Optics, Corning, NY, USA)) and used to build a library, started at 520 nm, unlike the spectroradiometer data shown in Figure 3c. Also, constant reflectance was not observed from 700 nm.

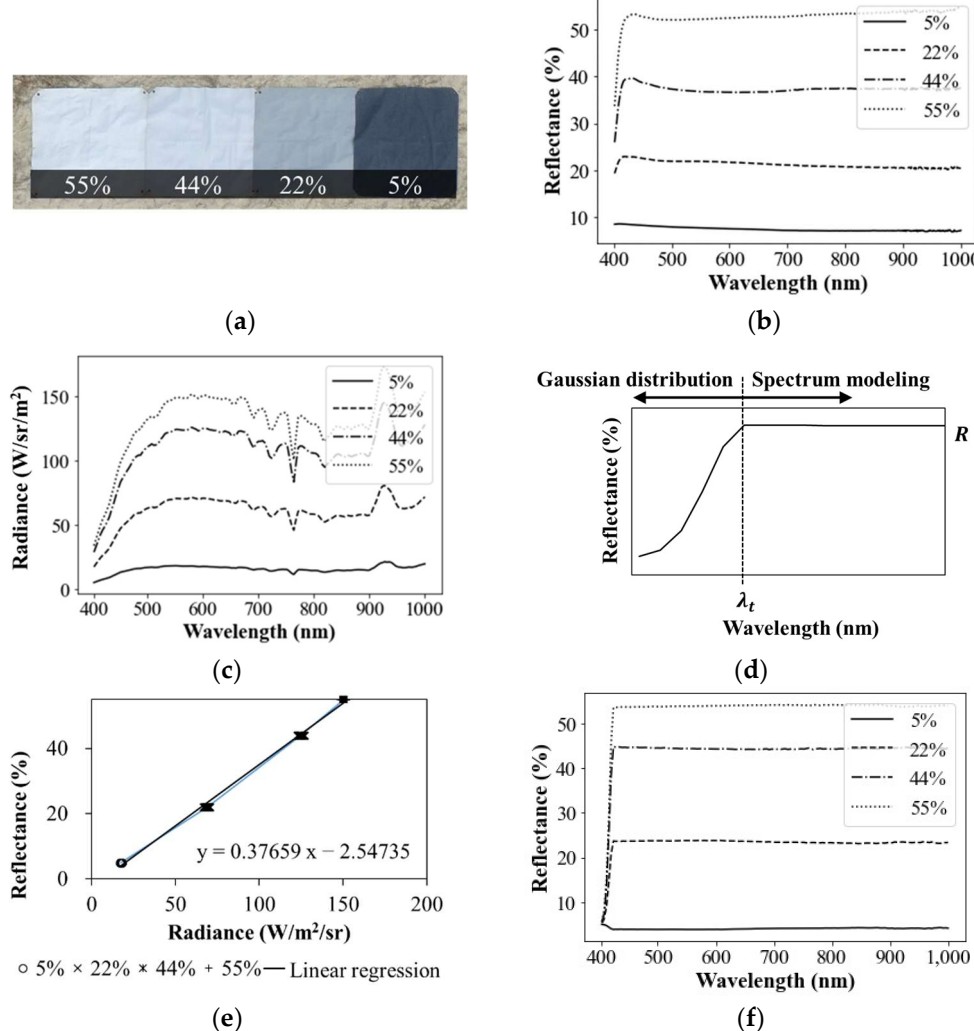

**Figure 3.** Derivation of radiometric calibration equation for a hyperspectral image using reflectance tarps. (**a**) The reflectance tarps; (**b**) spectrometer data from reflectance tarps; (**c**) hyperspectral sensor data from reflectance tarps; (**d**) concept of radiometric calibration; (**e**) radiance to reflectance rating at 597.66 nm; (**f**) calibrated reflectance tarps from hyperspectral sensor.

Therefore, reflectance modeling at 400–420 nm was based on the normal distribution (Equation (2)), by assuming the spectroradiometer data as the truth, so that similar trends are shown. For wavelengths above 420 nm, radiometric calibration was performed so that constant reflectance would be shown (Figure 3d).

$$N\left(x \vert \mu, \sigma^2\right) = \frac{1}{\left(2\pi\sigma^2\right)^{1/2}} \exp{-\frac{1}{2\sigma^2}(x-\mu)^2} \tag{2}$$

where $\mu$ is the mean and $\sigma^2$ is the variance. Since it must have the R-value at $\lambda_t$, $\frac{1}{\sqrt{2\pi\sigma^2}} \exp(\sigma) = R$ must be satisfied. In this study, $\sigma$ was calculated using the spectral

information of an actual spectrometer and applied to radiometric calibration, as shown in Figure 3d and Equation (3).

$$\begin{cases} N\left(xvert\lambda_t,\ \sigma^2\right) & \text{if) } \lambda < \lambda_t \\ \qquad R & \text{if) } \lambda \geq \lambda_t \end{cases} \qquad (3)$$

The above radiometric calibration, by building a specified radiance-reflectance relation, was performed for each individual 150 spectral bands collected from the hyperspectral sensor, respectively. The results are summarized in Table 2. Figure 3e shows an example of deriving the equation for converting radiance to reflectance at the wavelength 597.66 nm (No. 45, Table 2).

**Table 2.** The correlation coefficients between each 18 spectral samples and characteristic spectral libraries.

| No. | Chemicals | Correlation Coefficient | No. | Chemicals | Correlation Coefficient |
|---|---|---|---|---|---|
| 1 | 2-Chloroethanol | 0.871 | 10 | Hydrogen chloride | 0.910 |
| 2 | Glycidol | 0.926 | 11 | 2-Chloroethyldiethylammonium chloride | 0.934 |
| 3 | Methyl ethyl ketone | 0.989 | 12 | Thionyl chloride | 0.977 |
| 4 | Hydrogen fluoride | 0.948 | 13 | Phosphorus oxychloride | 0.912 |
| 5 | Bromine | 0.994 | 14 | Ammonium bifluoride | 0.861 |
| 6 | Arsenic trichloride | 0.814 | 15 | Toluene | 0.717 |
| 7 | Phosphorus trichloride | 0.881 | 16 | Sodium fluoride | 0.906 |
| 8 | Sodium cyanide | 0.948 | 17 | Potassium fluoride | 0.881 |
| 9 | Ammonia | 0.962 | 18 | Sulfuric acid | 0.977 |

Based on the radiometric calibration method presented aforehand, radiometric calibration was performed by applying radiance-reflectance relation established for each of the 150 bands in the entire captured collection of hyperspectral images (pixels). The complete process of the radiometric calibration was scripted using a dedicated software application for future use.

### 3.1.2. Extraction of Spectral Signatures

After applying the aforementioned radiometric calibration, spectral signatures rated as reflectance were extracted from the radiometrically calibrated hyperspectral images based on multiple pixels of the beaker holding the hazardous chemicals. Considering the outlier removal at a later step, more than 100 pixels were selected for the extraction of spectral information in the parts that are not affected by the scattered light due to the curved surface of the glass container (Figure 4a). This bundle of spectral signals will be averaged afterward to build a representative spectral signature for a given hazardous chemical. The extracted reflectance data pertained to the mixed samples (i.e., river water and chemicals), which has 2.5 cm depth of water in common. Herein, the spectral response could be differentiated to be commensurate with water depth. In this regard, we accordingly proposed and maintained a specified water depth for each container mentioned above to build a standardized spectral library. This way of consideration should be taken plausibly when certain soluble objects in liquid should be spectrally liberalized. In this study, a bundle of the spectral signatures derived from 100 pixels of the river water mixed with hydrofluoric acid image is shown in Figure 4b. As mentioned, we attempted to build a spectral library for overall 19 hazardous chemical samples, which were taken concurrently as shown in Figure 1e and enabled pulling out of the spectral signatures in an analogous environment. To facilitate this process, a dedicated in-house tool was developed to deal with a hyperspectral image, which basically allows the building of radiance-reflectance relations for each spectral band (Figure 4c,d) and simultaneously applies the relationships to the collected hyperspectral image. Subsequently, the software enabled the selection of multiple locations (i.e., 100 pixels) for each chemical container and the extraction of the spectral bundles (Figure 4a,b). The software was designed to efficiently load the heavy size of the hyperspectral image (2.4 GB); however, such large size tends to substantially slow down the conventional memory size of a personal computer. We resolved this by partially

loading specific bands or pulling out all band information for some designated pixels for a targeted analysis.

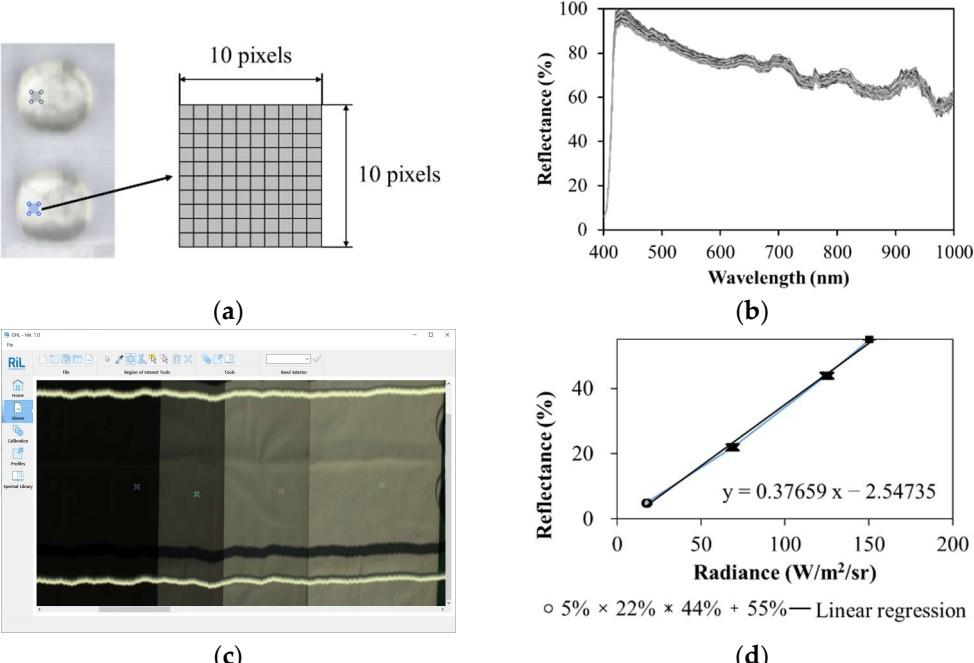

**Figure 4.** Extraction of spectral signatures from a hyperspectral image containing 19 chemicals using a mobile portable holder. (**a**) selection of specified pixels within a chemical container used as ingredient for building a spectral library for that chemical; (**b**) a bundle of hyperspectral signature extracted from selected pixels, where radiometric calibration was previously applied; (**c**) building radiance-reflectance relation using various reflectance tarps, where crossing point indicates a selected pixel corresponding to each reflectance percentile as well as raw radiance by the present hyperspectral camera (Corning microHSI); (**d**) a developed in-house software to make radiance-reflectance relation based on the previous step for conducting radiometric calibration and to apply the relations to entire bands and pixels.

### 3.1.3. Spectral Subtraction of Backgrounded Impacts

The radiance data of liquid collected through hyperspectral imaging includes base reflection, absorption by a water column, water surface reflection and absorption by the atmosphere. The equation for the total radiance is as follows:

$$L_T(\lambda) = L_b(\lambda) + L_c(\lambda) + L_s(\lambda) + L_p(\lambda) \simeq L_b(\lambda) \qquad (4)$$

where $L_T$ is the radiance; $\lambda$ is the wavelength; $L_b$ is the base reflection; $L_c$ is the water column absorption; $L_s$ is the water surface reflection; and $L_p$ is the atmospheric absorption. Bearing in mind that raw reflectance for chemical solvent river water encompassed several other backgrounded factors, spectral signatures were derived following the procedure described in Figure 3, which showed a considerably similar pattern irrespective of different types of chemicals. This output addresses the background impacts on spectral characteristics that exist inevitably for solvent materials and will differ for other cases accordingly with the specific spectral characteristics of river water. More specifically, the result shown in Figure 5a,b indicated that the background river impact was dominant to a much higher degree, rather than the chemicals. Consequently, the spectral signature of pure river water was eliminated in order to characterize the impact solely from solvent chemicals; otherwise the original spectral pattern would be inappropriate as the library was supposed to identify individual chemicals afterward. We assumed that the spectral signature for individual chemical will be differentiated to play a librarian role. In this regard, pure

river water without chemicals was also spectrally photographed and used to subtract from raw reflectance for each solvent chemical. Figure 5a,b exemplified a spectral signature of hydrogen fluoride in both conditions of mixed and pure river water, respectively, denoting that their spectral pattern looks remarkably analogous. When the spectra of pure water were subtracted from the mixed water, the residual spectral pattern became pronounced as shown in Figure 5c; this sort of residual spectral signature was identifiable and differentiable for each solvent chemical. The residual spectra can be negative when reflection of pure river water in certain spectral bands is higher than the solvent chemical. Collectively, the subtraction of backgrounded spectral impacts driven by river water was necessitated; subsequently, we adapted this background subtraction as a standardized step to build the spectral library of solvent chemicals.

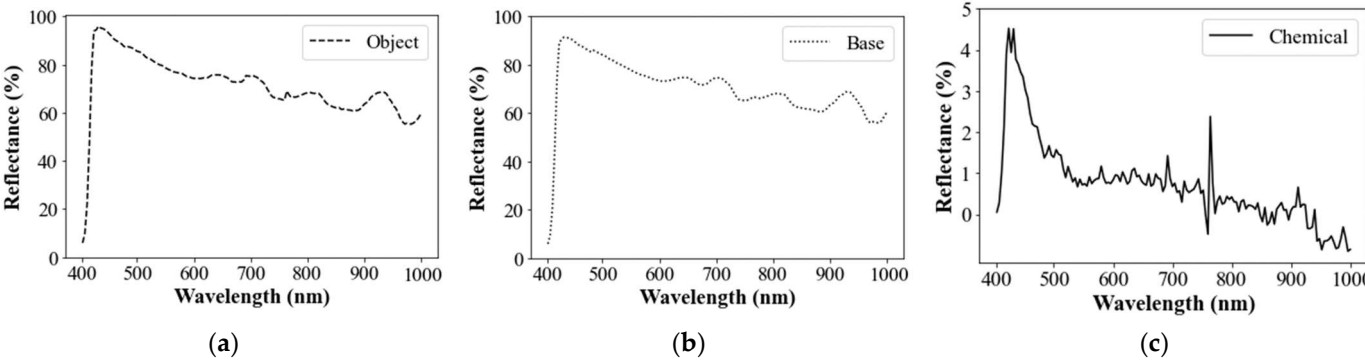

**Figure 5.** Derivation of spectral signature characterizing hazardous chemicals mixed with river water in terms of residual spectral pattern for solvent chemicals subtracted from a given pure river water. (**a**) Raw spectral signature exemplified for a solvent chemical mixed river water and hydrogen fluoride; (**b**) spectral signature of pure river water without any solvent chemicals; (**c**) residual spectral pattern for solvent hydrogen fluoride subtracting impact by pure river water.

### 3.1.4. Delineation of Mean Spectral Signature

A bundle of spectral signature using approximately 100 locations was taken from a spatially adjacent region as illustrated in Figure 6a. Interestingly, their resultant spectral patterns stemming from the identical chemical and monitoring environment demonstrated slightly different fluctuations, despite a similar main trend as described in Figure 6. Also, there exists spikes or outliers in spectral information, whose elimination is recommended. Therefore, it was necessary to select a method to eliminate outlier and then decompose a representative and mean spectral signature among a bundle of spectral signatures. There will be several useful methods to accomplish such a purpose. In this study, we applied a classic filtering algorithm to remove the spectral signatures, including the reflectance corresponding to outliers that leaves only the data within $\pm 3\sigma(\lambda)$, i.e., a 99.73% confidence interval of the reflectance values. To obtain the mean spectral value corresponding to each band, the Super Smoother method [20] was applied as one of the non-parametric regression methods as likely as loess or lowess [21,22]. The Super Smoother [23] computes the most probable value by applying weighted linear regression in each section after dividing bivariate data into elastic bandwidths and then determines a smoothed trend curve by estimating the most probable values, repeatedly in a given span, and connects them. Bandwidth is used to estimate the local pattern. If the bandwidth is too large, a flat curve close to a straight line is obtained (i.e., under-fitting); if it is too small, a curve with considerable bending of the regression function is obtained (i.e., overfitting). Therefore, a small bandwidth (i.e., optimal width size) was set to remove variations due to random behaviors and to reflect a local trend while maintaining the reflectance of each wavelength as much as possible. Such filtering and finding mean path algorithm of the super smoother was iterated, and finally, the overall trend after removing variations due to random behaviors of the collected data was obtained as shown in Figure 6b.

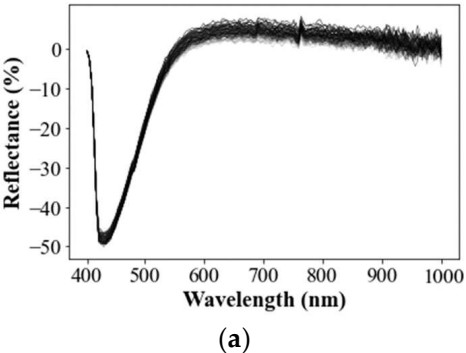
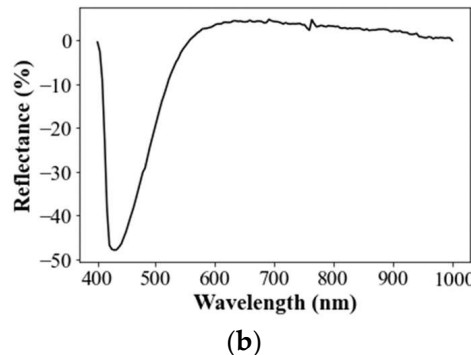

**Figure 6.** Derivation of a mean and filtered spectral signature for a hazardous chemical (bromine) using the super smoother. (**a**) A bundle of residual reflectance for bromine; (**b**) a derived spectral signature after applying filter and the super smoother.

### 3.1.5. Normalization of Mean Residual Reflectance

Spectral intensity responding to individual spectral band could differ according to in-situ conditions for the same solvent chemical, such as concentration of chemical, sunlight strength and type of hyperspectral camera. Although a major spectral factor of backgrounded river water was eliminated by the forced subtraction, the presence of such other remaining variants should be considered to build a generalized spectral library. This aspect is essential to diagnose an unknown solvent chemical based on the established library, but the new one could be possibly taken in different conditions. To address this, we normalized the residual spectral information using vector normalization with respect to total magnitude (I) cumulated by all residual reflectance as denoted in Equation (5).

$$R'(\lambda) = \frac{R(\lambda)}{I}, \ I = \sqrt{\int_{\lambda_{\min}}^{\lambda_{\max}} R(\lambda)^2} \tag{5}$$

$R'(\lambda)$ is the vector-normalized residual reflectance; $R(\lambda)$ is a raw residual reflectance; I is the magnitude cumulated for the vectorized residual reflectance; and $\lambda_{\max}$ and $\lambda_{\min}$ are the maximum and minimum wavelength of residual spectral signature, respectively. Figure 7 lists normalized residual spectral signatures for the designated 18 hazardous solvent chemicals. There could be distinctive features between chemicals, yet some are similar. For example, methyl ethyl ketone showed similar trend with Hydrogen fluoride but completely opposite compared with bromine. Hydrogen fluoride looks similar to Arsenic trichloride in the pattern, but their details in higher spectral band are different. On the other hand, sodium cyanide showed a unique feature that is very distinctive from other chemicals.

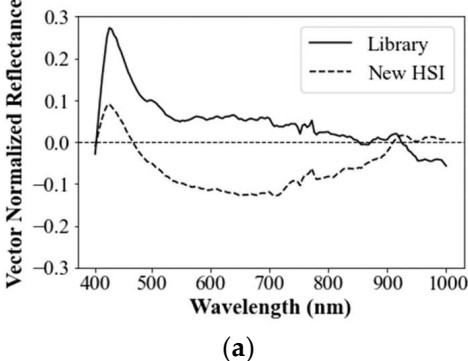
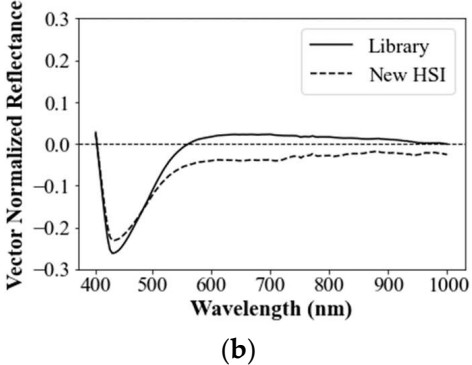

**Figure 7.** *Cont.*

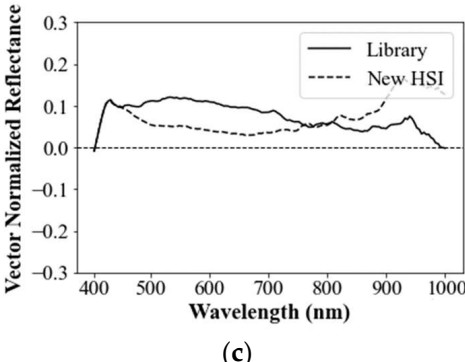
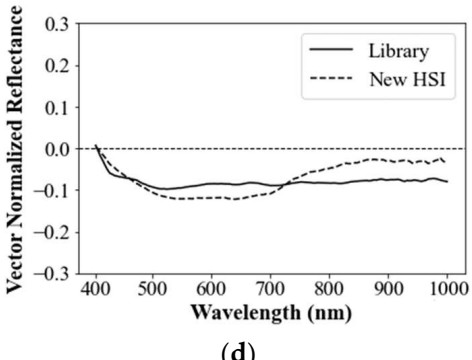

**Figure 7.** Exemplified comparison of the spectral spectrum between the spectral library and the HSI (Hyperspectral Image). (**a**) Hydrogen fluoride; (**b**) bromine; (**c**) phosphorus oxychloride; (**d**) ammonium bifluoride.

Spectral signature, such as that observed in normalized residual spectra in Figure 6, could have been immediately streamlined in a spectral library for identifying solvent chemicals, whereas full spectral signature has been generally used to identify and classify surface materials such as land cover [18–20]. Beyond using raw spectral information, we expected that various aforementioned steps to make raw spectral data more distinctive could be further augmented with the addition of backgrounded water spectra subtraction and normalization. However, the resultant normalized residual spectra can be prone to other submerged factors, like sunlight intensity; hence, this tentative spectral library should be validated to determine whether additional modification is necessary. In this regard, we conducted a preliminary diagnosis test, where the derived spectral library represented in Figure 7 was compared with newly captured hyperspectral signatures for 18 hazardous chemicals lagged with 2 h intervals where only sunlight strength was changed. Figure 7 demonstrates the feasibility of the developed library (Lib) in terms of sampled comparison for hydrogen fluoride, bromine, phosphorus oxychloride and ammonium bifluoride, respectively, where their spectral signatures were extracted from the new hyperspectral images (New HSI) in exactly the same way as described above.

The comparative results however indicated that, whereas bromine clearly showed highly analogous pattern and spectral value with new sample from qualitative perspective, hydrogen fluoride and ammonium bifluoride revealed less similarity in terms of spectral value, despite overall similar feature. Moreover, phosphorus oxychloride was hardly acceptable in terms of strong similarity. In fact, review of remaining chemicals did not show feasible agreement between developed library and new input. To evaluate quantitative comparison, the correlation coefficient between the two spectral spectrums was assessed. The correlation coefficient between the spectral library and the new HSI spectrum of each group was calculated for all the wavelength groups as follows:

$$r_{lib,new} = \frac{\sum_{i=1}^{n} \left\{ R(\lambda_{lib,i}) - \overline{R(\lambda_{lib})} \right\} \left\{ R(\lambda_{new,i}) - \overline{R(\lambda_{new})} \right\}}{(n-1)s_{lib}s_{new}} \qquad (6)$$

where $\overline{R(\lambda_{lib})}$ and $\overline{R(\lambda_{new})}$ are the mean reflectance of the spectral library and the new HSI spectrum, respectively; $s_{lib}$ and $s_{new}$ are the standard deviations of reflectance of the spectral library and the new HSI spectrum, respectively. In conjunction with prior qualitative view, bromine stood out with a high correlation coefficient of 0.979; however, correlation for most of the 17 hazardous chemicals showed highly poor performance of <0.6 and negatively correlated in other cases. For example, correlations for ammonium bifluoride and hydrogen fluoride were 0.63 and 0.26, respectively. Phosphorus oxychloride resulted in a negative correlation of −0.44. This may be attributed to the fact that only bromine has chromaticity, while other chemicals were mostly invisible, which can prominently account

for the superior performance of bromine. The results suggested further modification of the spectral library.

### 3.1.6. Characteristics of Spectral Library

When the whole spectrum was applied as a library, as shown in Figure 7, we found that the spectral bands irrelevant to reflective characteristics of chemical could be randomly different according to spectral nature and ambient measurement conditions. These irrelevant spectral bands negatively affected the correlation assessment against reference in high degree, thus should be better excluded from the spectral library. Subsequently, it would be optimal to identify unique bands that enable to characterize specific chemicals. In fact, the approach of solely using optimal bands (not whole) has been widely accepted in various hyperspectral applications, such as NDVI for vegetation [24], NDWI for water [25] and optimal band ratio analysis (OBRA) for submerged bathymetry [8]. In this study, however, we did not develop a specialized index or finding optimal band ratio similarly with former studies. Rather, we selected specific band groups among whole spectrum to characterize a specific chemical as likely as a signature. We presume that such signatures should discriminate each other within the targeted chemicals. This is an important perspective and also a limitation since our objective was to only identify a chemical among the given pool of hazardous chemicals, where concept of ranking was adapted to prioritize candidates. We referred to this specific signature of chemical as "characteristic spectral library for hazardous chemicals", which was built on top of a previously processed spectrum following the aforementioned procedure.

To derive such characteristic library, various plausible ways can be used for further manipulation of the given processed library. As a top principle, it is necessary to extract common and unique parts from multiple spectrums acquired on various measurement conditions. In this way, the greater number of spectrums will better pull-out uniqueness of the spectral reflectance, where data mining techniques, such as machine learning, will work in certain degree to delineate specific patterns. Nevertheless, spectral samples in the presence of hazardous chemicals in the natural rivers are especially hard to obtain compared with other conventional natural objects. The safe way of handling chemicals was to bring such river water and artificially dissolve and mix them in a controlled experiment.

In this study, therefore, we followed a simple correlation defined in Equation (6) and preliminarily applied available only two different spectrums to manually pull-out common characteristics, though it would not be sufficiently generic and automated.

More specifically, in the two spectrums captured per each chemical, we defined 146 pairs, each having approximately 20 nm of coverage and applied the entire wavelength range of 400–1000 nm with 150 spectral bands. Then the correlation coefficient was assessed for each pair and ranked in order of highest value. For example, phosphorus trichloride was sorted in top five highly correlated pairs of 689.76–709.79, 633.7–653.72, 469.52–485.54, 509.56–525.58 and 733.81–785.87 nm, where the correlation coefficients in those pairs were 0.932, 0.909, 0.881, 0.834 and 0.711, respectively (Figure 8a). In addition, we attempted to further move the average cascadedly starting from the first two (i.e., 0.932 and 0.909) down to the fifth rank until the highest mean correlation coefficient was acquired.

As a result, phosphorus revealed that the first two bands had the highest moving average (Figure 9b), which characterized phosphorus in the spectral library. Herein, the number of characteristic coverages can be 2–5. Given that the number of maximum characteristic coverage of 5 was arbitrarily chosen, it is sufficient to characterize the uniqueness and similarity but can be more. Figure 9 shows signatured coverage to represent spectral characteristic of all 18 solvent chemicals, where multiple spans of bands were adapted to characterize the chemicals. For example, glycidol pertained five coverages as a characteristic spectral signature. An examination of the characteristic spectral libraries indicated that all 18 chemicals exhibited different characterized spectral properties. In particular, bromine, light brown in color, revealed significantly lower reflectance than other chemicals until 500 nm at which it turned blue. Interestingly, the overall shape of the spectrum of

bromine was similar to that of sodium cyanide, having no chromaticity cyanide especially in the lower reflectance. Similarly, fluorine compounds, i.e., sodium fluoride and potassium fluoride, showed similar overall spectral patterns. In addition, 2-chloroethyl, glycidol, hydrogen fluoride, bromine, phosphorous trichloride, 2-chloroethyl-diethyl ammonium chloride, thionyl chloride, sodium fluoride and potassium fluoride showed similar patterns in the wavelength range of 749.83–773.86 nm.

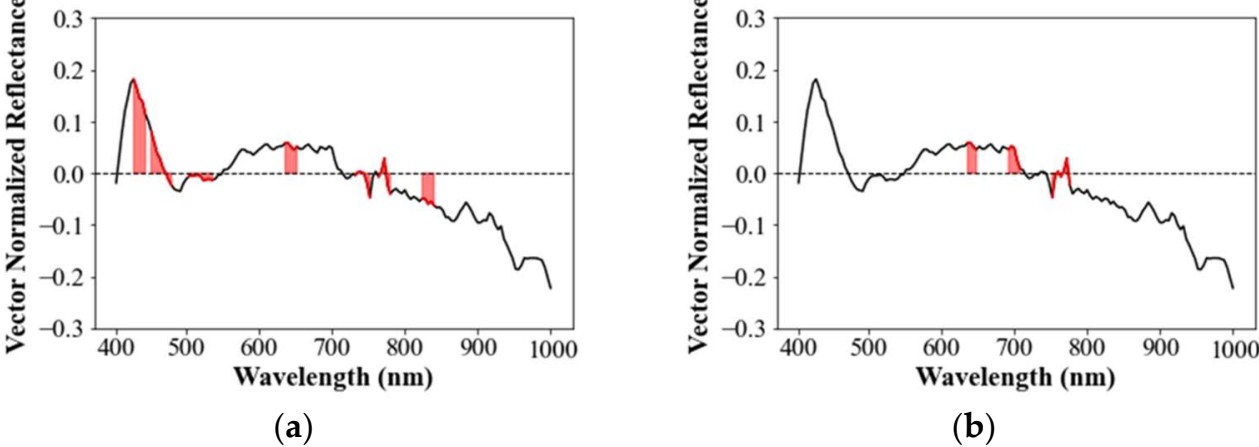

(**a**)    (**b**)

**Figure 8.** Characteristic spectral library of phosphorus trichloride (red color). (**a**) Primary characteristic spectral library of Phosphorus trichloride; (**b**) final characteristic spectral library of phosphorus trichloride.

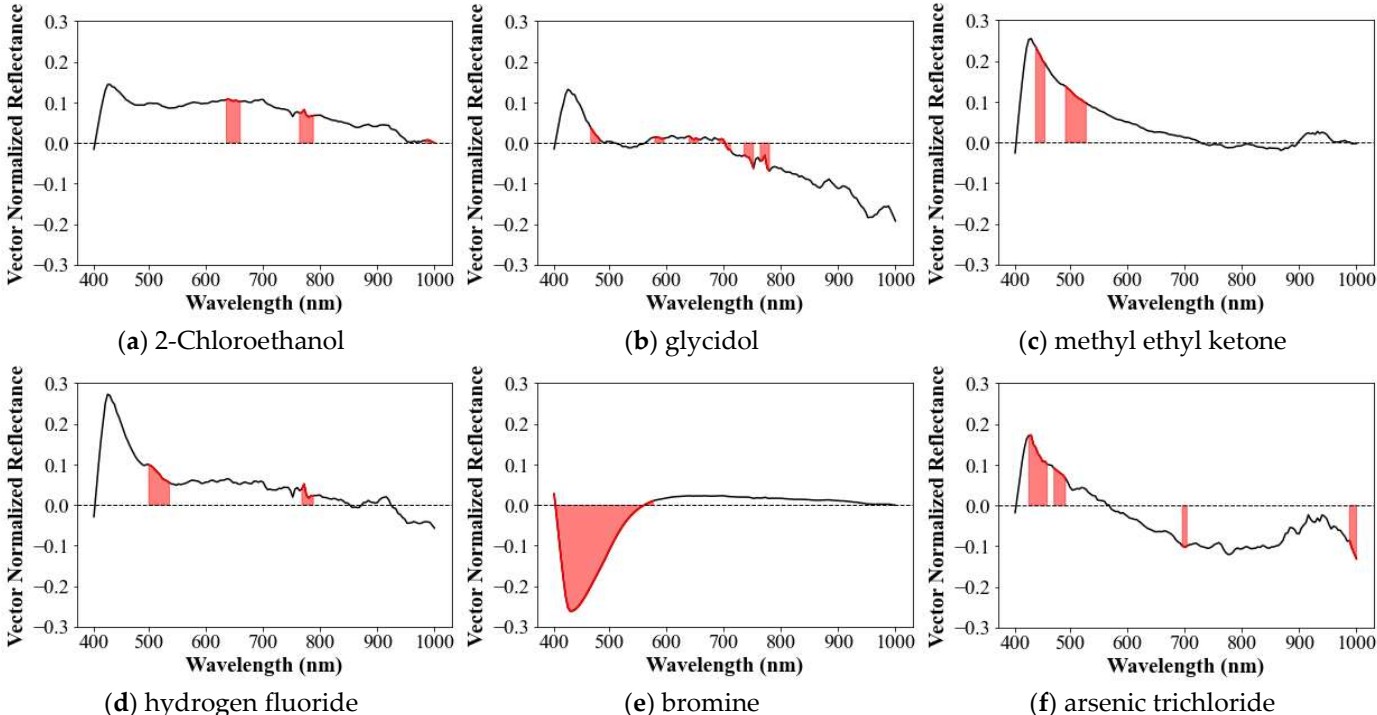

(**a**) 2-Chloroethanol    (**b**) glycidol    (**c**) methyl ethyl ketone

(**d**) hydrogen fluoride    (**e**) bromine    (**f**) arsenic trichloride

**Figure 9.** *Cont.*

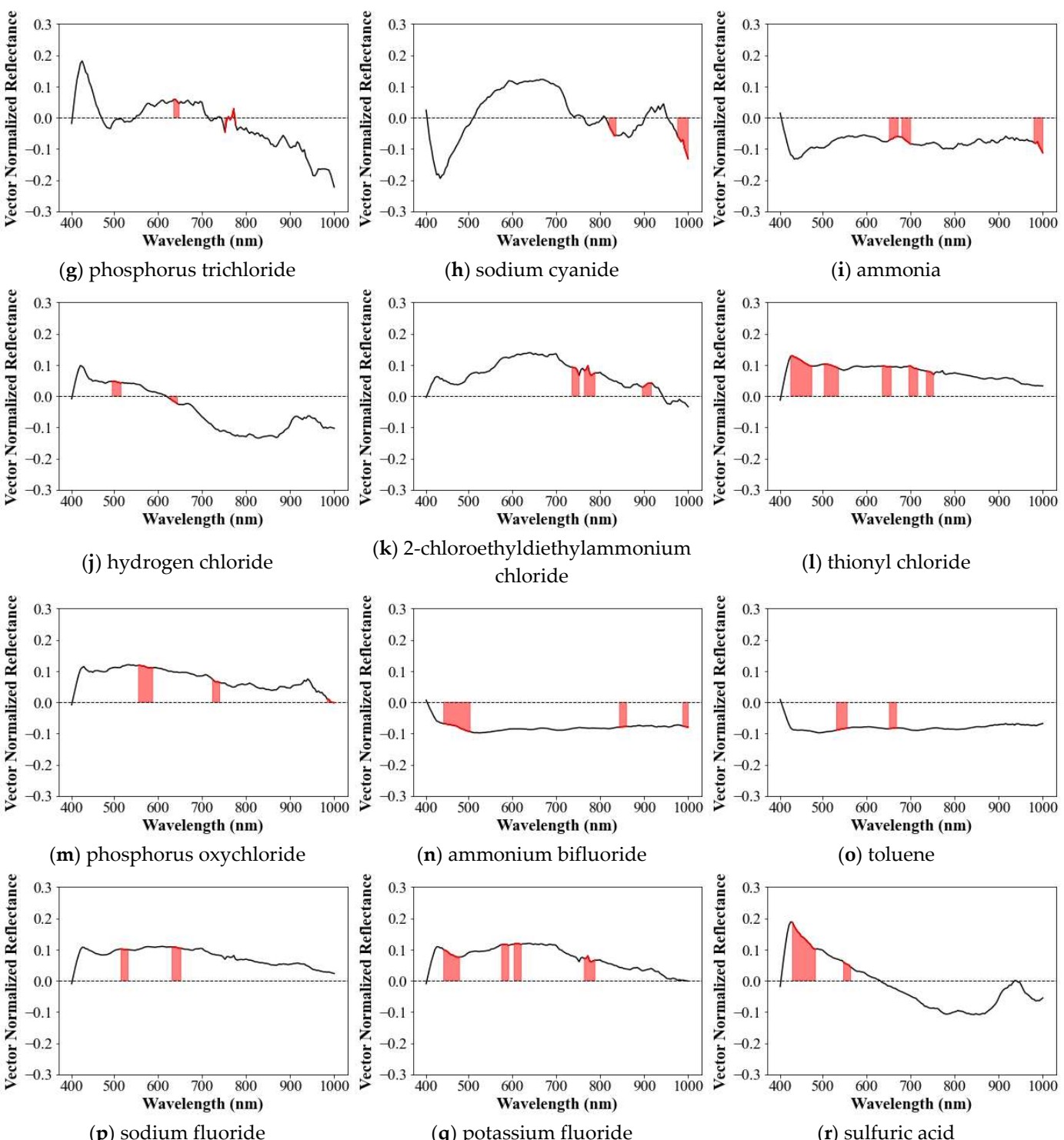

**Figure 9.** Characteristic spectral library. (**a**) 2-Chloroethanol; (**b**) glycidol; (**c**) methyl ethyl ketone; (**d**) hydrogen fluoride; (**e**) bromine; (**f**) arsenic trichloride; (**g**) phosphorus trichloride; (**h**) sodium cyanide; (**i**) ammonia; (**j**) hydrogen chloride; (**k**) 2-chloroethyldiethylammonium chloride; (**l**) thionyl chloride; (**m**) phosphorus oxychloride; (**n**) ammonium bifluoride; (**o**) toluene; (**p**) sodium fluoride; (**q**) potassium fluoride; (**r**) sulfuric acid.

Basically, the strategy of present study imposed several characteristic coverages as signature in case certain chemicals have different overall patterns, which is anticipated to better characterize and recognize a specified one. For the chemicals with similar patterns after ap-

plying steps of vector normalization, however, this strategy does not consistently guarantee high recognition between them. For example, the characteristic library of bromine can work suitably with that similar spectral coverage of sodium cyanide; thus, recognition process will yield to sodium cyanide as the best match rather than bromine that can be a secondary feasible match. We postulated that above drawback is unavoidable. Consequently, we suggested further recognition guideline to recommend more candidates (e.g., within ranking 3) in addition to the most similar one, where correlation coefficient between new and library account for the plausible criteria as a sort of probability. In fact, this approach is also useful in practice since we can substantially narrow down and specify the type of chemicals and track nearby spilling locations, such as chemical factory if chemical database was priorly established (e.g., among chemical factories located in upstream of Gam Stream in this study). Despite the different number of coverages for each chemical and certain degree of similarity, the resulted characteristic spectral library in Figure 9 is useful to identify and classify the type of leaked chemical.

Noting that only two spectral resources in an analogous environment (e.g., 2 h difference) were ingested, these characteristic spectral libraries are not certainly conclusive, thus limitedly applicable for identifying a new unknown chemical. Nevertheless, the results demonstrated that the derived spectral library well documented their distinctive features, and proof and good example of applying proposed procedure, indicating that the more spectral resources from various environment and variable are added, the better consolidated spectral characteristics of chemicals will be obtainable. In Figure 10, we summarized an end-to-end step to build a characteristic spectral library.

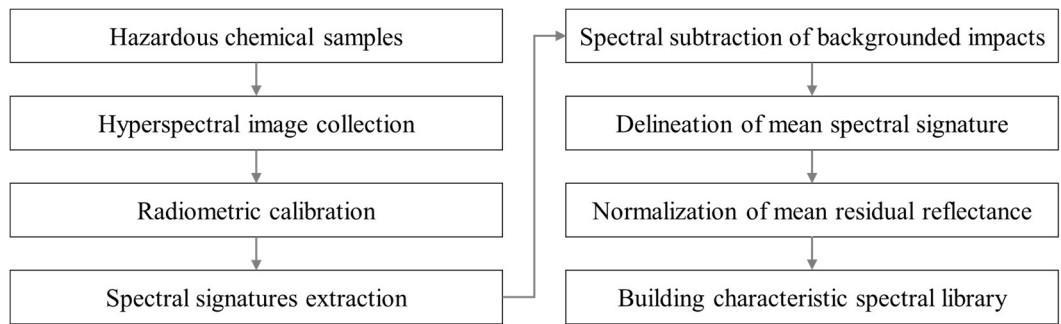

**Figure 10.** A flow chart to describe an end-to-end step for building characteristic spectral library for hazardous chemicals proposed solvent in rivers.

### 3.2. Recognition Process

Given that a spectral library for 18 solvent hazardous chemicals was established, we demonstrated how the library reliably recognizes a new unknown chemical (Figure 11). The unknown spectral information will be compared with the librarized chemicals one by one and ranked accordingly with the computed correlation. Note that whole recognition process only occurs among the given spectral library and mandatorily suggests suitable candidates within ranking 3 by referencing correlation coefficient, rather than assertively designating a single one. We herein exemplified chemical recognition process using newly photographed hyperspectral images for each hazardous chemical. For each chemical, spectral information was extracted and normalized after excluding ambient water as illustrated so far. Such manipulated spectral information was partially utilized up to characteristic spectral coverages specific to librarized chemicals. We prepared the same 18 samples of chemicals as in the library and took hyperspectral images in the slightly different environment when used during library construction. The unknown chemicals were hyperspectrally imaged at around 6 pm with the other conditions being identical. During sampling, the light intensity was lower than when imaging for constructing the spectral library. Then we individually applied the recognition procedure for 18 unknown chemicals and tested how feasibly the library yielded right candidates.

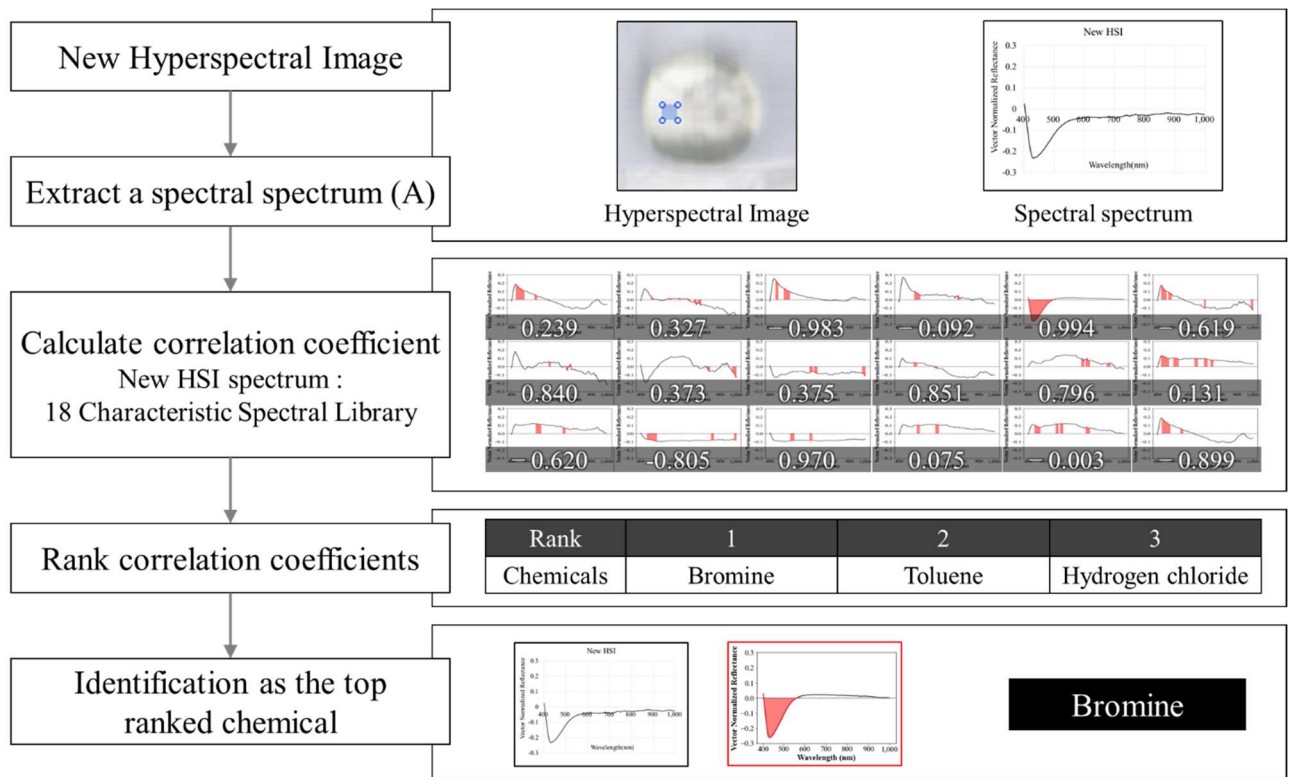

**Figure 11.** Chemicals recognition concept using a characteristic spectral library.

The correlation coefficients between each of 18 spectral samples and characteristic spectral libraries were assessed in Table 2 and ranked accordingly with correlation and simultaneously provided the rank to fit with each sample in Table 3. For example, the library recognized 2-chloroethanol correctly as first rank, but arsenic trichloride was ranked thirdly. In particular, bromine was firstly ranked in spite of having significantly lower reflectance than chemicals until 500 nm at which it turned blue. Low concentrations of chemicals with chromaticity could be better ranked though most chemicals in the library were transparent. We proposed that, when the rank of a chemical was within third, it became conceded as the library worked by narrowing down candidates for the unknown chemical. The greater number of chemicals in the library corresponds to higher accuracy of capturing the type of chemicals. Overall, the recognition rate showed that the developed library gave precisely correct answers for more than one-half of the samples (10 samples) as first ranked, and a total of 13 chemicals were among the top three chemicals (i.e., the recognition rate of 72.2%). Table 2 demonstrated the correlations between 18 spectral samples and a spectral library, which led to more than 0.8 except one sample (e.g., Toluene), implying an analogous nature of the derived library. In low rank cases, Table 3 imposed that practitioners narrow down possible candidates of leaked chemicals. This recognition rate indicated that our efforts to adapt a characterizing process substantially enhanced the success rate, considering that prior to using the characteristic library (meaning that full spectrum was used as signature to assess correlation), the recognition rate was approximately 53.3%. Nevertheless, five chemicals could not reach within the third rank, such that a characteristic library missed matching them. They were mostly transparent solvent subordinate of sodium and hydrogen type. In particular, toluene was completely out of ranking scope (ranked tenth), though this chemical was slightly visible as floating texture when it was mixed with water like spilled oil. The score of unmatched 27.8% of samples (i.e., beyond third rank) implied that, though we considered and standardized all different environmental conditions and furthermore adapted the characteristic concept, there might be other factors that were not considered.

**Table 3.** Evaluation of recognition rate—Example identification using a characteristic spectral library.

| Rank | Number | Chemicals |
| --- | --- | --- |
| 1 | 10 | 2-Chloroethanol, glycidol, methyl ethyl ketone, hydrogen fluoride, bromine, ammonia, thionyl chloride, ammonium bifluoride, potassium fluoride, and sulfuric acid |
| 2 | 2 | 2-Chloroethyldiethylammonium chloride and phosphorus oxychloride |
| 3 | 1 | Arsenic trichloride |
| 4 | 2 | Hydrogen chloride and Sodium fluoride |
| 5 | 1 | Sodium cyanide |
| 6 | 1 | Phosphorus trichloride |
| 10 | 1 | Toluene |

In addition, the present chemical samples were only taken at difference in light intensity. In general, chemical samples can be measured in other higher order of difference in the measurement equipment (i.e., spectroradiometer or hyperspectral sensor), platform (terrestrial or airborne), weather, chemical concentration, different type of ambient river water, the bed materials (sand, gravel, mud, vegetated) and the container for holding samples. In those cases, the recognition success rate can be lower. Unfortunately, given the difficulties of securing hazardous chemicals in those conditions, it was not possible to obtain diverse cases of samples and was limited to developing a more solid spectral library for sampling chemicals, which was an unavoidable limitation of handling hazardous chemicals. Nevertheless, given such limited conditions, we suggest that the derived characteristic library work in certain acceptable range and can enhance the recognition rate. Also, considering previous similar studies, the developed library in this study was measured on condition where air correction was ignored; thus, those spectral images from low altitude UAV and terrestrial hyperspectral system can be eligible.

## 4. Conclusions

In this study, we developed a strategy for building a standard spectral library by characterizing hazardous chemicals using a hyperspectral sensor. Upon recognizing that various limitations occur when the spectral library is directly applied to the identification of chemical, the characteristic spectral library reflected several distinctive characteristic wavelength bands for each chemical and used them to better recognize an unknown chemical. Summarized procedures for building a characteristic spectral library are outlined here: (a) assuming that hazardous chemical samples have the analogous concentration, hyperspectral images are captured per container with identical hyperspectral sensor in operation and surrounding imaging conditions; (b) the collected spectral information is converted into reflectance through radiometric calibration where the spectral signatures of the sampled chemicals are extracted from the calibrated hyperspectral images; (c) spectral subtraction of backgrounded impact stemmed particularly from ambient river water; (d) outliers are removed from the extracted data and checked for validity, then mean spectral signature is delineated using a smoothing method (e.g., super smoother); (e) each spectrum per chemical is normalized for further standardized correlation assessment; (f) characteristic features from the normalized full spectrum are extracted to uniquely distinguish the spectral signature of a chemical. Based on the above procedure, we have developed a characteristic library for 18 hazardous chemicals assuming a scenario of leaking them into rivers and tested the feasibility of the library to recognize unknown chemicals, which meaningfully worked with >70% success rate.

Throughout developing a standardized procedure to develop a spectral library and perform a recognition test, we have the following conclusions and suggestions for future research:

(1) Hyperspectral-based detection of solvent hazardous chemicals in the riverine environment was scarce, thus very little former research was referable. In this regard, though our initial effort proposed in this study is preliminary and should be more

verified, such a library will be referenceable for further research. Practically, the library can be useful for the real-time detection and distinction between hazardous chemicals by facilitating a response to the occurrence of chemical accidents. We highly anticipate that more subtle aspects of enhancing performance of the spectral library will continue to be investigated. From this perspective, further research efforts as well as appropriate guidelines to synthesize various water environment indices such as the concentration of organic matters, concentration of algae, turbidity and water depth are required.

(2) At the initial stages of this study, we considered more than 100 hazardous chemicals, but only covering a small portion of them given the danger of handling them since several of them were forbidden to be treated without permission from national security. In addition, practical abundance to fulfill a solid library even for available chemicals was therefore limited, because it was impossible to obtain sufficient in-situ samples. Instead, the mobile carrier mounting hyperspectral camera was useful for mimicking line-scanning hyperspectral imaging process in outdoor hyperspectral sampling of very low concentration of hazardous chemical samples.

(3) We arbitrarily assigned a criterion for the maximum number of characteristic bands (or coverage) as five. Similarly, ranking criteria to define recognition success was set up as less than third rank. Though we verified that these criteria are suitable to build and utilize a spectral library, such ambiguity in properly defining the optimal criteria remained and should be resolved. For example, we speculate that spectral libraries can better work if developed with the greater number of spectral images, reflecting diverse measurement environments with applying advanced AI techniques to train the capacity to recognize chemicals' signature. Also, the rank in the recognition process was provided in this study as correlation coefficient, rather than considered to suggest probability.

(4) Characterizing distinctive features among the full spectrum was effective in addition to excluding the ambient spectral impact, such as river water, and normalizing to reduce the impacts from concentration. However, other factors influencing spectral signature should be considered.

(5) Beyond identifying hazardous chemicals, the approach proposed herein for building a characteristic spectral library can also be applied for recognizing other types of solvent materials.

(6) For the better recognition rate and practical uses, various water environment indices, such as the concentration, weather conditions, turbidity, water depth and water color, should be considered ensuring they contain a great number of in-situ spectral images. In those cases, artificial intelligence techniques might be a good solution.

**Author Contributions:** Conceptualization, D.K. and Y.D.K.; methodology, H.Y.; software, H.Y.; validation, Y.G. and S.-H.N.; formal analysis, Y.G.; investigation, Y.G. and D.K.; resources, Y.D.K.; data curation, S.-H.N. and Y.G.; writing—original draft preparation, Y.G.; writing—review and editing, D.K.; visualization, Y.G.; supervision, D.K. and Y.D.K.; project administration, Y.D.K.; funding acquisition, Y.D.K. and D.K. All authors have read and agreed to the published version of the manuscript.

**Funding:** Korea Agency for Infrastructure Technology Advancement (KAIA) grant funded by Korea Ministry of Environment (MOE) (Grant 22DPIW-C153746-04).

**Institutional Review Board Statement:** Not applicable.

**Informed Consent Statement:** Not applicable.

**Data Availability Statement:** Not applicable.

**Acknowledgments:** This research was supported by the Korea Agency for Infrastructure Technology Advancement (KAIA) grant funded by Korea Ministry of Environment (MOE) (Grant 22DPIW-C153746-04).

**Conflicts of Interest:** The authors declare no conflict of interest.

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
