# Peer review of "A Standardized Procedure to Build a Spectral Library for Hazardous Chemicals Mixed in River Flow Using Hyperspectral Image"

_remotesensing, doi:10.3390/rs15020477_

Round 1

Reviewer 1 Report

The authors proposed a standardized procedure to build a spectral library for hazardous chemicals. The topic is generally interesting, but the experiments and results should be improved. Please see below my comments.

1. The current work is a qualitative identification method, however,the efficiency of the library to recognize unknown chemicals under ideal conditions is not very good.

2. Various water environment indices, such as the concentration, weather conditions, turbidity, water depth, water color ……, should be considered to create a practical library, ensuring it contains a great number of spectral images. AI techniques might be a good solution to this problem.

3. In Fig 10, why the correlation coefficient 0.985 is out of the rank?

Author Response

The review report to respond reviewer's comments is attached. 

Reviewer 2 Report

· The authors describe the process of spectral library creating for detection of the hazardous chemicals mixed in river flow. They use hyperspectral images from UAV. The authors collected a sufficient number of chemicals samples and formed a spectral library on it.

· The authors use UAVs to obtain hyperspectral images for river water quality assessment. Compared to traditionally used remote sensing data, this approach can give more accurate results due to more flexible initial setup of equipment and libraries for processing the received data.

· Methodology of the spectral library creation is well described. In the introduction it is necessary to expand the part describing related works in the field of hyperspectral images and UAV usage in water quality assessment.

·  In conclusion, the methodology of the study is briefly presented and the results obtained are listed.

· Most of the citations were a bit old. Please refer the recent publications.

· In section 5 I would like to see a comparison table for recognition rates of different samples, chemicals, library versions, etc. Numerical information is easier to understand in the table form than in the plain text. 

Author Response

(The authors gave the same response as above.)

Reviewer 3 Report

The authors propose a referable standard procedure to build a spectral library for 18 types of hazardous chemicals mixed in river based on pre-scanning hyperspectral sensors. The materials and method were well described. In general, the manuscript is well written and can be a valuable reference to the related readers.

There is only one suggestion for the author. Adding an overall data processing flow chart for building this spectral library will make the manuscript easier to read.

Author Response

(The authors gave the same response as above.)

Round 2

Reviewer 1 Report

The author have made some replies to the last round of comments, but not comprehensive. Please reply to the last round of comments in more detail. According to the reply of this round, the reviewer acknowledges the author's modification and improvement. If the reply is further improved, it is acceptable.

Author Response

Authors added our note in the attached file.